

# Effects of three weeks base training at moderate simulated altitude with or without hypoxic residence on exercise capacity and physiological adaptations in well-trained male runners

Longyan Yi[1], Jian Wu[2], Bing Yan[1], Yang Wang[1], Menghui Zou[3], Yimin Zhang[1,4], Feifei Li[5], Junqiang Qiu[6] and Olivier Girard[7]

[1] China Institute of Sport and Health Sciences, Beijing Sport University, Beijing, China
[2] School of Kinesiology and Health, Capital University of Physical Education and Sports, Beijing, China
[3] China Athletics School, Beijing Sport University, Beijing, China
[4] Key Laboratory of Exercise and Physical Fitness (Beijing Sport University), Ministry of Education, Beijing, China
[5] Centre for Health and Exercise Science Research, Department of Sport, Physical Education and Health, Hong Kong Baptist University, Beijing, China
[6] Department of Exercise Biochemistry, Exercise Science School, Beijing Sport University, Beijing, China
[7] School of Human Sciences (Exercise and Sport Science), University of Western Australia, Perth, Western Australia.

Corresponding authors
Feifei Li, lifeifei.daisy@hotmail.com
Junqiang Qiu, qiujunqiang@bsu.edu.cn

## ABSTRACT

**Objectives:** To test the hypothesis that 'live high-base train high-interval train low' (HiHiLo) altitude training, compared to 'live low-train high' (LoHi), yields greater benefits on performance and physiological adaptations.

**Methods:** Sixteen young male middle-distance runners (age, 17.0 ± 1.5 y; body mass, 58.8 ± 4.9 kg; body height, 176.3 ± 4.3 cm; training years, 3–5 y; training distance per week, 30–60 km.wk$^{-1}$) with a peak oxygen uptake averaging ~65 ml.min$^{-1}$.kg$^{-1}$ trained in a normobaric hypoxia chamber (simulated altitude of ~2,500 m, monitored by heart rate ~170 bpm; thrice weekly) for 3 weeks. During this period, the HiHiLo group (n = 8) stayed in normobaric hypoxia (at ~2,800 m; 10 h.day$^{-1}$), while the LoHi group (n = 8) resided near sea level. Before and immediately after the intervention, peak oxygen uptake and exercise-induced arterial hypoxemia responses (incremental cycle test) as well as running performance and time-domain heart rate variability (5-km time trial) were assessed. Hematological variables were monitored at baseline and on days 1, 7, 14 and 21 during the intervention.

**Results:** Peak oxygen uptake and running performance did not differ before and after the intervention in either group (all $P > 0.05$). Exercise-induced arterial hypoxemia responses, measured both at submaximal (240 W) and maximal loads during the incremental test, and log-transformed root mean square of successive R-R intervals during the 4-min post-run recovery period, did not change (all $P > 0.05$). Hematocrit, mean reticulocyte absolute count and reticulocyte percentage increased above baseline levels on day 21 of the intervention (all $P < 0.001$), irrespective of group.

**Conclusions:** Well-trained runners undertaking base training at moderate simulated altitude for 3 weeks, with or without hypoxic residence, showed no performance improvement, also with unchanged time-domain heart rate variability and exercise-induced arterial hypoxemia responses.

## KEY POINTS OF THE PAPER

1) Undertaking base training at moderate simulated altitude for three weeks, with or without hypoxic residence, did not confer any performance or physiological benefits among well-trained young runners.

2) The significant within-athlete variability observed in this study emphasizes the importance of shifting focus away from seeking universal 'best' answers and towards exploring individual-specific altitude training solutions.

3) Well-trained runners engaged in base training at moderate simulated altitude for three weeks, whether with or without hypoxic residence, experienced no improvement in performance. Additionally, time-domain heart rate variability and exercise-induced arterial hypoxemia responses remained unchanged.

## INTRODUCTION

Since the 1968 Mexico City Olympics, athletes and coaches have routinely incorporated altitude training into their seasonal plans (*Mujika, Sharma & Stellingwerff, 2019*). Noticing the dominance of Eastern African runners, many quickly adopted the classical altitude training method known as 'live high-train high' by travelling to moderate altitude locations (1,800–2,200 m) such as Colorado Springs, Font Romeu, and St Moritz. The prevailing paradigm was that a decrease in alveolar oxygen partial pressure with increasing altitude, due to reduced atmospheric pressure, would trigger heightened erythrocyte production, leading to increased peak oxygen uptake ($\dot{V}O_{2peak}$) and eventually improved performance (*Saunders et al., 2013*; *Płoszczyca, Langfort & Czuba, 2018*). While remaining popular, 'live high-train high' interventions have received criticism due to their cost, the inherent decline in training quality, and logistical problems with such training camps (*Álvarez-Herms et al., 2015*).

Other popular altitude training approaches include 'live high-train low' and 'live low-train high' (as known as LoHi), or a combination of both (*Millet et al., 2010*). An updated nomenclature for altitude training methods, describing opportunities to combine different modalities, has been proposed by *Girard, Brocherie & Millet (2017)*. A promising mixed-method regimen, that may allow for better maintenance of training intensity in reference to 'live high-train low', is known as 'live high-base train high-interval

train low' (HiHiLo). As originally described by *Chapman, Stray-Gundersen & Levine (1998)*, this paradigm requires athletes to reside and perform low-intensity training at moderate altitude (~2,500 m) for several weeks, while high-intensity (interval) training is conducted at lower elevations (~1,250 m). In elite runners, a 3% increase in $\dot{V}O_{2peak}$, accompanied by improvement of several hematological variables (*e.g.*, hemoglobin mass and hematocrit), resulted in faster (−5.8 s) 3-km running time after a 4-week HiHiLo protocol (*Stray-Gundersen, Chapman & Levine, 2001*). Contrastingly, a 3-week HiHiLo intervention did not improve aerobic capacity estimated by $\dot{V}O_{2peak}$ despite significantly higher values of hematological variables in elite biathletes (*Czuba et al., 2014*). More substantial evidence is required to support the effectiveness of the HiHiLo intervention, as athletes and coaches may be hesitant to adopt this altitude training protocol without stronger confirmation of its benefits.

Screening responses, such as exercise-induced arterial hypoxemia (EIAH), may provide an indication to the degree to which the athlete responds to training or competition, either in normoxia or hypoxia (*Chapman et al., 2011*). Individuals demonstrating an arterial oxygen saturation ($SpO_2$) less than 92% during sea-level exercise, already on the shoulder of the oxyhemoglobin dissociation curve, likely experience larger declines in $\dot{V}O_{2peak}$ at altitude compared with those who maintain $SpO_2$ at higher levels (*Chapman, 2013*). Outside hematological markers, fewer studies have determined whether or not chronic hypoxia improves EIAH and/or heart rate variability (HRV) measurements, and conflicting results have often been reported; for instance, an increase (*Povea et al., 2005*) or not (*Bernardi et al., 2001*) in the low frequency component of HRV. In a group of ten elite triathletes, those who exhibited greater performance gains in 800-m swim time trials at sea level also demonstrated a shift towards greater vagal predominance. This change was observed on both the first and last day of a 20-day 'live high-train low' altitude training camp at 1,650 m (*Hamlin et al., 2011*). Additionally, *Marshall et al. (2008)* found that 90-min intermittent hypoxia at a clamped $SpO_2$ of 80% for ten consecutive days, compared to a placebo control, reduces EIAH severity at exhaustion during an incremental exercise from 91% before to 94% after the treatment. While intermittent hypoxia (at rest) may induce some positive physiological adaptations at the muscle tissue level, no direct (*i.e.*, same participants) comparison exists of the effects of HiHiLo and LoHi on EIAH at both submaximal and maximal exercise intensities and HRV indices.

The aim of this study was to determine the effects of 3-week simulated 'live high-base train high-interval train low' (HiHiLo) altitude training, compared to 'live low-train high' (LoHi), on changes in physical performance, selected hematological variables, as well as EIAH and HRV at sea level immediately after returning from the camp.

## METHODS

### Participants

A convenience sample of sixteen lowland male middle-distance runners, all from the same local club, was recruited for the study. Participants were classified as 'Trained/Developmental' (Tier 2) using established criteria (*McKay et al., 2022*), taking ~6-d training per week and one session each day, which were composed of running from 5,000

**Table 1 Physical performance assessment parameters.**

| | Baseline characteristics | |
| --- | --- | --- |
| | HiHiLo ($n = 8$) | LoHi ($n = 8$) |
| Age (years) | 16.9 ± 1.5 | 17.0 ± 1.5 |
| Body mass (kg) | 58.8 ± 5.5 | 58.7 ± 4.3 |
| Height (cm) | 177.5 ± 3.7 | 175.1 ± 4.9 |
| 5,000-m time (min) | 17.3 ± 1.0 | 17.4 ± 2.4 |
| $\dot{V}O_{2peak}$ (ml.kg$^{-1}$.min$^{-1}$) | 64.4 ± 6.1 | 66.6 ± 6.6 |
| $\dot{V}O_2$ of 240 W (ml.kg$^{-1}$.min$^{-1}$) | 53.3 ± 9.9 | 56.5 ± 7.3 |
| Time point when EIAH (min) | 13.3 ± 3.6 | 13.9 ± 3.0 |
| Workload when EIAH (W) | 228.8 ± 27.5 | 234.4 ± 32.0 |
| SpO$_2$ of 240 W (%) | 91.1 ± 3.9 | 93.6 ± 2.8 |
| Minimal SpO$_2$ (%) | 89.9 ± 2.9 | 90.6 ± 3.1 |

**Note:**
$\dot{V}O_{2peak}$, peak oxygen uptake; EIAH, exercise-induced arterial hypoxemia; SpO$_2$, arterial oxygen saturation.

to 10,000 m generally. All participants were near sea-level residents and had not travelled to altitude (>1,500 m) or engaged in simulated altitude exposure in the 6 months prior to the study. They provided written informed consent following a detailed explanation of all experimental procedures and possible risks. The study was approved by the Sport Science Experimental Ethics Committee for Human Subjects of Beijing Sport University (Reference No. 2022012H).

## Study design

The experimental design comprised the following phases: a pre-intervention period (pre-tests) at sea level, during which physical performance (5-km time trial and $\dot{V}O_{2peak}$) and baseline testing (BL) for hematological parameters was conducted; a 3-week period (*Intervention*) when participants were randomly assigned to a 'live high-base train high-interval train low' (HiHiLo; $n = 8$) or a 'live low-train high' (LoHi; $n = 8$) according to their fitness level; and a post-intervention period (post-tests) for physical performance assessment (Table 1). Performance tests were conducted on two consecutive days immediately after the intervention, in an invariant order (*i.e.*, randomized across participants but held constant for each individual at each time point within ±2 h), under similar temperate conditions. All participants were familiar with testing procedures as part of their regular physical performance assessment. Although not recorded, particular attention was paid to food intake, hydration, and sleep habits during the experiment, ensuring that participants from all groups received similar diets and adhered to consistent bedtime schedules. In all cases, participants were asked to replicate their last meals, while avoiding alcohol and caffeine intakes during the 24 h preceding each test scheduled in the same time slot. Tap water was provided *ad libitum*. Participants were vigorously encouraged during all efforts (*Yan et al., 2021*).
## Altitude exposure

During *Intervention*, HiHiLo participants resided 10 h per day in normobaric hypoxic bedrooms, equipped with an automatic controlled centralized hypoxia system (Lowoxygen systems; GmbH, Nussdorf-Traunstein, Germany), with a room temperature of ~24 °C (*Yan et al., 2021*). The carbon dioxide level was maintained under 3,000 ppm, and regularly tested using a sensor every 10 min. The system ventilated the room air, diluted the air with 99% nitrogen, and circulated the room air six times per hour. It automatically adjusted the flows of nitrogen and fresh air to control the oxygen concentration at the pre-set value of ~2,800 m ($FiO_2$ 14.8%). LoHi participants also resided 10 h per day at sea level in the same bedrooms, but without turning on the hypoxic system. All participants were blinded to the actual hypoxic level.

Both groups followed the same daytime training program, involving 3 weekly 60-min exercise routines. The exercise intensity was monitored by maintaining a heart rate of ~170 bpm or ensuring $SpO_2$ remained below 92%. The training was conducted at a simulated altitude of ~2,500 m ($FiO_2$ 15.4%). The exercise protocol included a general 5-min warm-up, followed by a 30-min cycling (Ergomedic 828E; Monark, Vansbro, Sweden) and 30-min running on the treadmill (h/p/ cosmos Mercury 4.0; h/p/cosmos sports & medical gmbh, Nussdorf-Traunstein, Germany). All training sessions were overseen by the main coach of the running club. Participants were instructed not to engage in any other forms of structured training during the duration of the study.

## Incremental test

A graded exercise test protocol was performed on a cycle ergometer (Ergomedic 839E; Monark, Vansbro, Sweden) to determine $\dot{V}O_{2peak}$. The test started with cycling at 120 W, with increments of +30 W every 3 min until exhaustion. Gas exchange parameters were measured using a mix chamber metabolic cart system (Max-I; Physio-Dyne Instrument, Quogue, NY, USA). The flow, oxygen and carbon dioxide sensors were calibrated each day before testing following manufacturer's instruction. The gas samples were averaged every 15 s, and the highest values for $\dot{V}O_2$ over 15 s was regarded as $\dot{V}O_{2peak}$. Four criteria were used to determine maximal efforts: (1) a plateau or levelling off in $\dot{V}O_2$, defined as an increase of less than 1.5 ml.min$^{-1}$.kg$^{-1}$ despite progressive increases in exercise intensity; (2) a final respiratory exchange ratio of 1.1 or above; (3) a final heart rate above 95% of the age-related maximum; (4) volitional exhaustion. EIAH was determined when arterial oxygen saturation (measured at fingertips) (3100; Nonin Medical INC, Plymouth, MN, USA) decreased by 4% from the resting level (*Dempsey et al., 2006*; *Yan et al., 2021*).

## Athletic performance

Running performance was evaluated through a 5-km time trial performed on a 400-m outdoor synthetic track at sea level. The starts were individualized in a time-trial format to mitigate group or pacing effects. All trials were completed between 4 and 6 pm.

## Hematological parameters

Venous blood samples were collected by venous punctures pre-intervention at BL and on days 1, 7, 14, and, 21 during *Intervention*. All samples (5 ml) were drawn by trained phlebotomists from an antecubital forearm vein into an EDTA vacutainer, with the participants in the sitting position, between 7 and 9 am under fasting condition. Hematological parameters measured included red blood cell count (RBC), hemoglobin concentration (Hb), hematocrit (Hct) (KX-21N; SYSMEX, Hyogo, Japan) and reticulocytes (ADVIA-120; BAYER, Tokyo, Japan).

## Heart rate variability

Heart rate, and time-domain HRV indexes (the average R-R interval duration in a measurement (mean RR interval) and root mean square of the successive differences RMSSD) were taken (Polar RS800CX; Polar Electro Oy, Kempele, Finland) during the 5-km time trial (run) and every minute during the ensuing 4-min recovery period (+1, +2, +3, and +4 min post-run) in a standing position. Data were processed, stored, and downloaded using Polar ProTrainer-5™ professional training software. Because all HRV indexes were not normally distributed (Kolmogorov-Smirnov test), LnRMSSD was calculated as the natural logarithm of root mean square of the successive differences (*Shaffer & Ginsberg, 2017*).

## Statistical analysis

Values are expressed as mean ± SD. Two-way repeated-measures ANOVAs (camp (BL *vs.* days 1, 7, 14 and/or 21) or time (pre- *vs.* post-tests) × group (HiHiLo *vs.* LoHi)) were used to compare incremental test and athletic performance, or haematological data. Three-way repeated-measures ANOVAs (time (pre- *vs.* post-tests) × group (HiHiLo *vs.* LoHi) × exercise (run *vs.* +1, +2, +3 and +4 min post-run)) were used to compare HRV data. Normal distribution of the residuals was tested using the Kolmogorov–Smirnov test. To assess assumptions of variance, Mauchly's test of sphericity was performed using all ANOVA results. A Greenhouse–Geisser correction was applied to adjust the degree of freedom if an assumption was violated, while *post hoc* test pairwise-comparisons with a Bonferroni-adjusted *P* values were performed if a significant main effect was observed. Partial eta-squared ($\eta^2$, with $\eta^2 \geq 0.06$ representing a *moderate* effect and $\eta^2 \geq 0.14$ a *large* effect) values were calculated. All statistical calculations were performed using SPSS statistical software V.27.0 (IBM Corp., Armonk, NY, USA). Statistical significance was set at $P < 0.05$ (*Girard, Mariotti-Nesurini & Malatesta, 2022*).

# RESULTS

## Performance and exercise capacity

From pre- to post-tests, the 5-km running time did not change significantly for both HiHiLo (1,038 ± 55 *vs.* 1,017 ± 42 s; −2.1 ± 1.2%) and LoHi (1,052 ± 144 *vs.* 1,037 ± 119 s; −1.4 ± 2.5%) (Fig. 1A). There were no significant time (all $P > 0.107$) or group (all $P > 0.269$) main effects, and no significant time × group interaction (all $P > 0.353$) for all variables derived from the incremental test (Figs. 1B to 1F).

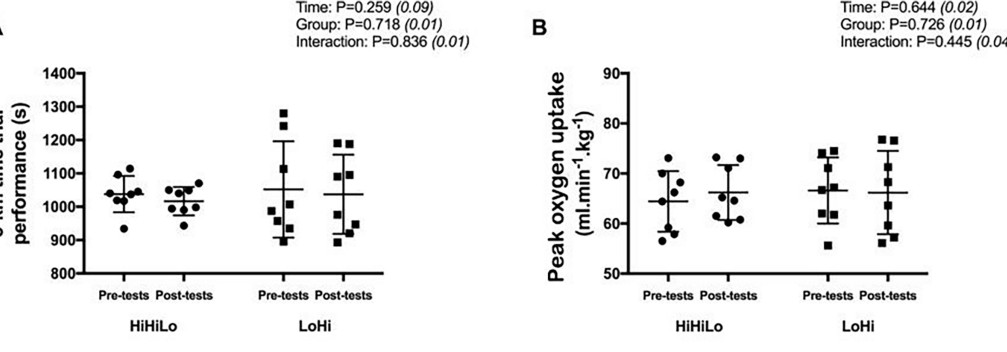

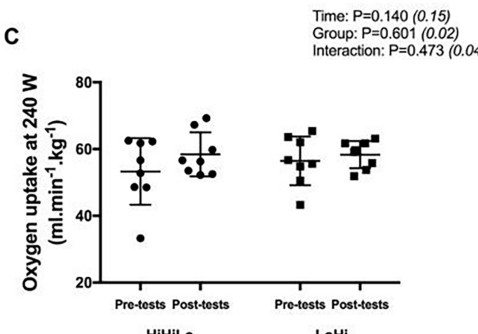
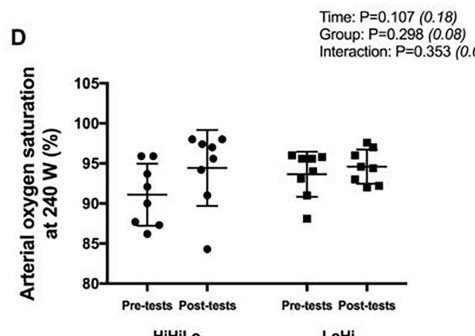

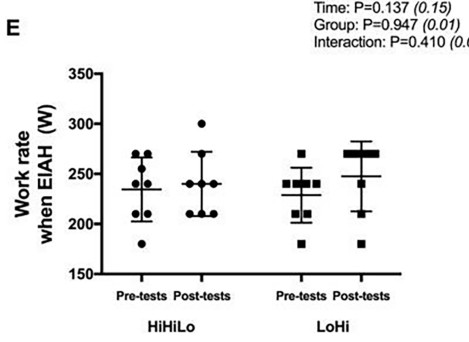
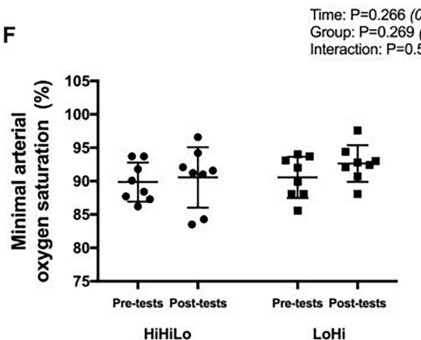

**Figure 1 Performance measures obtained before (pre-tests) and after (post-tests) 'live high-base train high-interval train low' (HiHiLo) and 'live low-train high' (LoHi) altitude training.** Values are mean ± SD. ANOVA main effects of time, group and interaction are stated along with partial-eta squared into brackets.

Table 2 Changes of hematological parameters before and 1st 7th 14th, and 21st day during intervention.

| | HiHiLo (n = 8) | | | | | LoHi (n = 8) | | | | | P (Interaction, Time, Group) |
|---|---|---|---|---|---|---|---|---|---|---|---|
| | Baseline | 1st d | 7th d | 14th d | 21st d | Baseline | 1st d | 7th d | 14th d | 21st d | |
| RBC | 4.9 ± 0.3 | – | 4.8 ± 0.6 | 4.9 ± 0.3 | 5.1 ± 0.3 | 4.9 ± 0.3 | – | 5.1 ± 0.2 | 4.8 ± 0.3 | 5.0 ± 0.4 | 0.240, 0.163, 0.931 |
| HB | 153.6 ± 8.9 | – | 148.5 ± 16.4 | 152.9 ± 8.4 | 157.6 ± 9.8 | 151.5 ± 6.6 | – | 155.3 ± 7.4 | 147.6 ± 7.2 | 155.1 ± 10.5 | 0.421, 0.070, 0.825 |
| HCT (%) | 41 ± 3 | – | 40 ± 5 | 40 ± 2 | 42 ± 3 | 40 ± 2 | – | 41 ± 2 | 39 ± 2 | 42 ± 3 | 0.300, 0.066, 0.805 |
| Retic# | 0.05 ± 0.01 | 0.07 ± 0.01* | 0.08 ± 0.02* | 0.06 ± 0.01* | 0.07 ± 0.01* | 0.05 ± 0.01 | 0.06 ± 0.02* | 0.07 ± 0.01* | 0.06 ± 0.02* | 0.06 ± 0.02* | 0.413; 0.000; 0.104 |
| Retic% | 1.13 ± 0.16 | 1.52 ± 0.17* | 1.70 ± 0.29* | 1.41 ± 0.15* | 1.51 ± 0.23* | 1.03 ± 0.28 | 1.28 ± 0.43* | 1.40 ± 0.27* | 1.24 ± 0.41* | 1.30 ± 0.45* | 0.757; 0.000; 0.106 |
| L-Retic% | 92.2 ± 2.1 | 91.0 ± 1.2 | 92.1 ± 2.2 | 90.5 ± 1.7 | 92.6 ± 2.6 | 92.3 ± 3.4 | 91.5 ± 3.1 | 92.8 ± 1.9 | 93.2 ± 3.5 | 93.4 ± 2.5 | 0.283; 0.100; 0.357 |
| M-Retic% | 7.15 ± 2.09 | 8.32 ± 1.88 | 7.18 ± 2.10 | 8.83 ± 1.54# | 6.94 ± 2.58 | 6.78 ± 3.08 | 7.70 ± 3.01 | 6.50 ± 2.05# | 5.86 ± 3.32# | 5.86 ± 2.47 | 0.377; 0.045; 0.263 |
| H-Retic% | 0.67 ± 0.37 | 0.64 ± 0.25 | 0.75 ± 0.33 | 0.68 ± 0.33 | 0.43 ± 0.11 | 0.98 ± 0.51 | 0.81 ± 0.26 | 0.68 ± 0.29 | 0.93 ± 0.51 | 0.77 ± 0.45 | 0.448; 0.371; 0.064 |
| CHr | 37.9 ± 1.3 | 37.8 ± 1.2 | 37.6 ± 1.2 | 37.3 ± 1.3 | 37.2 ± 1.8 | 36.6 ± 1.1 | 37.0 ± 1.0 | 37.0 ± 0.7 | 36.8 ± 0.9 | 36.9 ± 0.9 | 0.579; 0.718; 0.172 |
| CHm | 35.7 ± 0.7 | 35.4 ± 0.7*# | 35.5 ± 0.7*# | 35.3 ± 0.7# | 35.4 ± 0.8# | 33.6 ± 1.2 | 33.6 ± 1.1# | 33.7 ± 1.3# | 33.5 ± 1.1# | 33.7 ± 1.1# | 0.673; 0.006; 0.002 |

Notes:
HiHiLo, live high-base train high-interval train low; LoHi, live low-train high; RBC, red blood cell; HB, hemoglobin; HCT, hematocrit; Retic#, Reticulocyte absolute count; Retic%, Reticulocyte percentage; L-Retic%, low fluorescence reticulocyte percentage; M-Retic%, medium fluorescent reticulocyte percentage; H-Retic%, high fluorescence reticulocyte percentage; CHr, cellular hemoglobin in reticulocytes; CHm, percentagecellular hemoglobin in mature red blood cells.
* Significantly different from pre-intervention.
# Significant difference between groups.

## Blood parameters

Table 2 lists all hematological parameters. Mean (*i.e.*, pooled values for the two groups) reticulocyte absolute count (Day 1: +31.4 ± 7.2%, Day 7: +47.4 ± 11.1%, Day 14: +24.3 ± 7.8% and Day 21: +32.6 ± 12.5%) and reticulocyte percentage (Day 1: +30.3 ± 7.4%, Day 7: +46.5 ± 10.6%, Day 14: +24.3 ± 9.0% and Day 21: +33.1 ± 14.2%) increased above BL levels during the intervention (all $P < 0.001$), irrespective of group. Overall, low fluorescence reticulocyte percentage and medium fluorescence reticulocyte percentage were significantly higher (+1.9 ± 0.7%, $P = 0.027$) and lower (−19.5 ± 7.6%, $P = 0.045$) at Day 21 in reference to Day 1.

There were no significant time ($P = 0.069$ and $P = 0.070$) or group ($P = 0.931$ and $P = 0.825$) main effects, and no significant time × group interaction ($P = 0.358$ and $P = 0.421$) for red blood cells (Fig. 2A) and hemoglobin (Fig. 2B). Hematocrit increased significantly from BL to Day 21 (+4.0 ± 2.3%, $P = 0.006$), irrespectively of group (Fig. 2C).

## Heart rate variability

Compared to during the run, heart rate decreased progressively each minute during the 4-min post-run recovery period (Fig. 3A), while mean RR intervals lengthened (Fig. 3B). No changed occurred for LnRMSSD (Fig. 3C).

# DISCUSSION

## Performance and exercise capacity

Neither group demonstrated significant improvement in $\dot{V}O_{2peak}$ or 5-km running time post-intervention. Contrastingly, *Robertson et al. (2010)* reported that 3 weeks of HiHiLo (at ~2,200 m, 4 d.wk$^{-1}$) led to substantial (~5%) $\dot{V}O_{2peak}$ gains in middle-distance runners with comparable aerobic fitness level, and a smaller improvement in 3-km running time. These authors further reported that a LoHi group had ~2% increase in $\dot{V}O_{2peak}$, yet with no

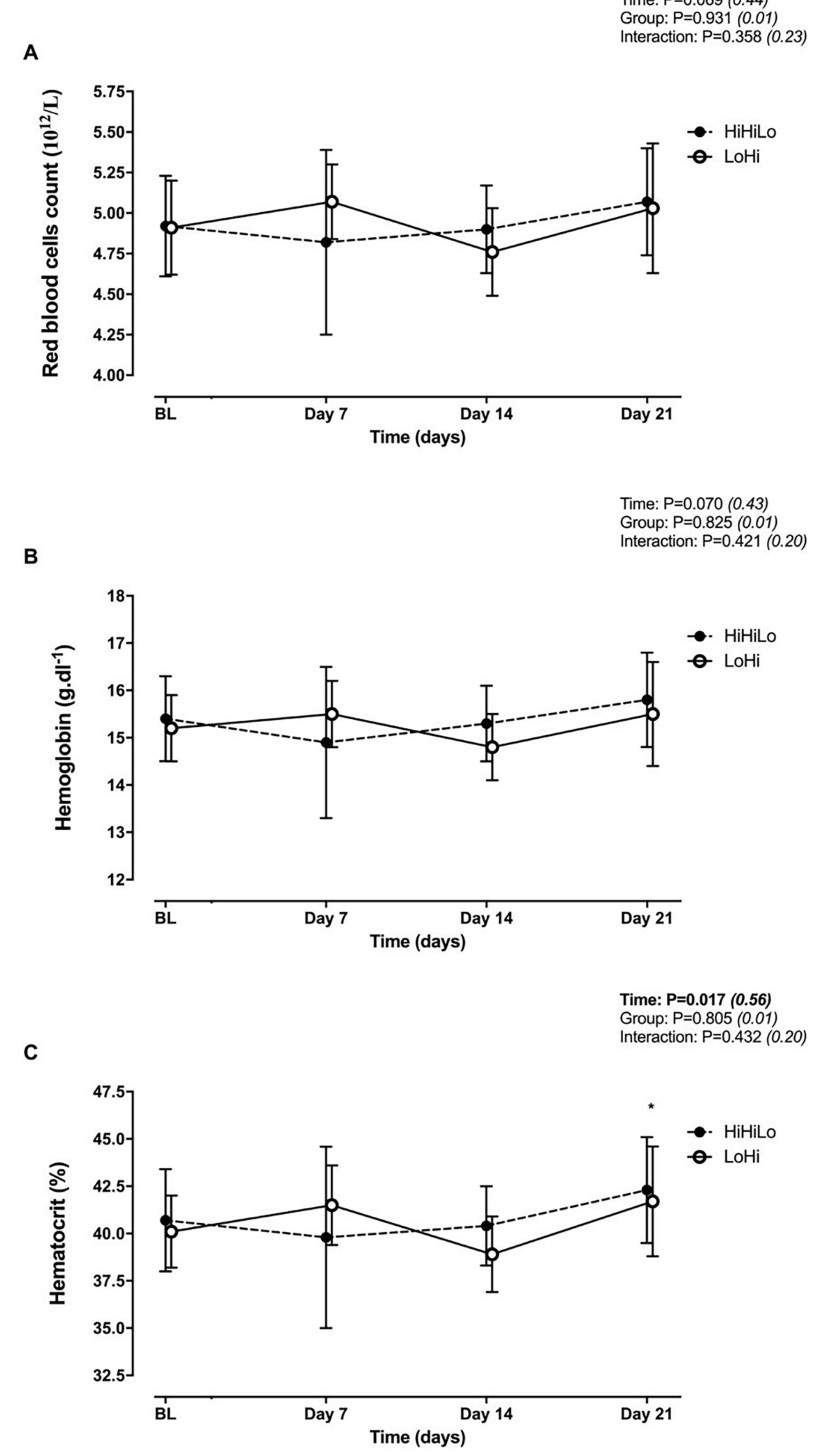

**Figure 2 Changes in red blood cells (A), hemoglobin (B) and hematocrit (C) for 'live high-base train high-interval train low' (HiHiLo) and 'live low-train high' (LoHi) groups.** Values are mean ± SD. Data were collected at baseline (BL) and days 7, 14 and 21 into a 3-wks training camp. Participants were randomly allocated to either a 'live high-base train high-interval train low' (HiHiLo, $n = 8$) or 'live low-train high' (LoHi, $n = 8$) group. ANOVA main effects of time, group and interaction are stated along with partial-eta squared into brackets. An asterisk (*) indicates significant difference from BL ($P < 0.05$).                             

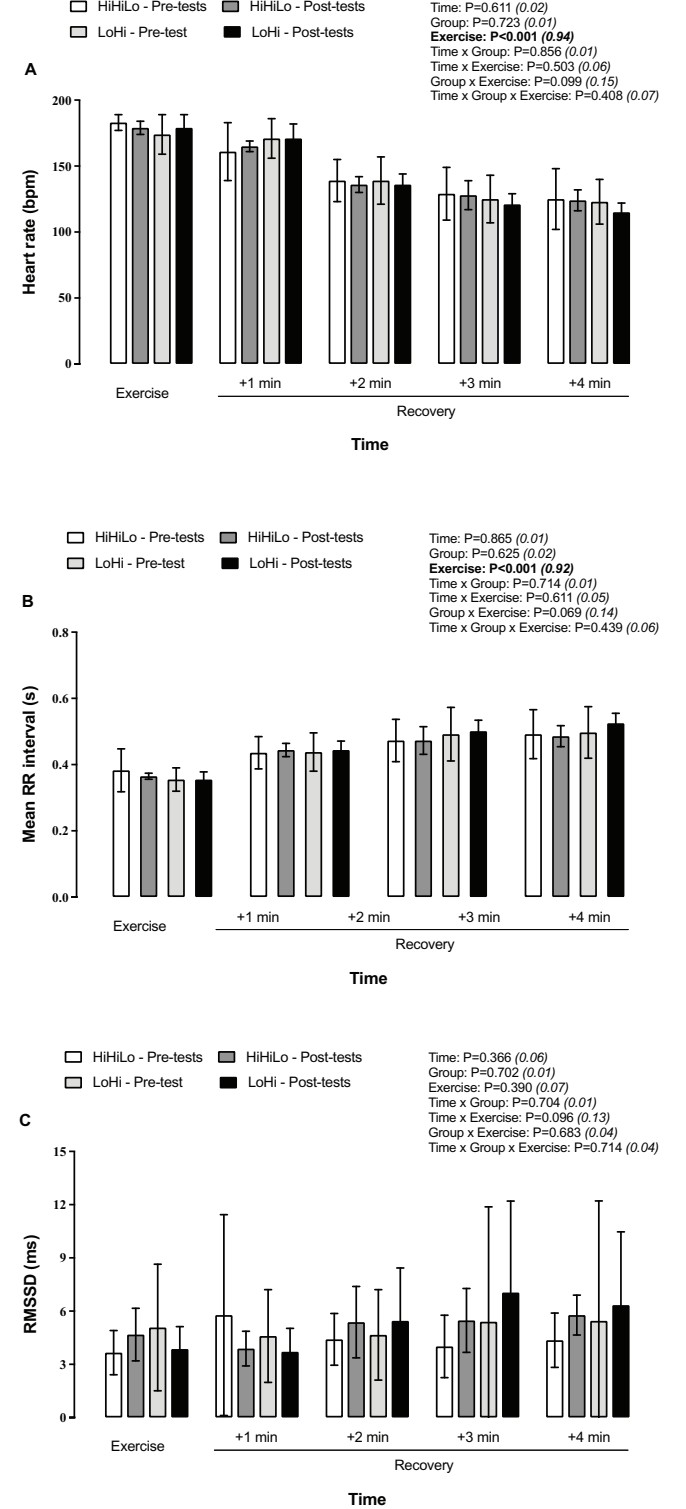

**Figure 3 Heart rate (A), mean RR interval (B) and LnRMSSD (C) during a 5-km running time trial and the 4-min post-run recovery period.** Participants were randomly allocated to either a 'live high-base train high-interval train low' (HiHiLo, $n = 8$) or 'live low-train high' (LoHi, $n = 8$) group. ANOVA main effects of time, group, exercise and interactions between these factors are stated along with partial-eta squared into brackets.

corresponding improvement in time trial performance. However, other researchers failed to report improvement in $\dot{V}O_{2peak}$ and/or time trial performance immediately post-intervention using HiHiLo (*Czuba et al., 2014*; *Rodriguez et al., 2015*) and LoHi (*Bonetti & Hopkins, 2009*) interventions. Discrepancies may arise from differences in daily hypoxic exposure, severity and/or nature of the stimulus characterizing the 'altitude dose', and training background of tested individuals. Using a method for calculating the total hypoxic dose (*Garvican-Lewis, Sharpe & Gore, 2016*), which considers both the total time under hypoxia and the level of hypoxia (*i.e.*, the kilometer hours, calculated as km.h = (elevation above sea level/1,000) × hours of exposure), the hypoxic dose received by the altitude training group in this study (~610 km.h) was relatively similar to doses implemented by others (*e.g.*, 500–950 km.h; *Robertson et al., 2010*; *Saugy et al., 2014*; *Brocherie et al., 2015*). Although attractive, a metric based on the magnitude of the stimulus (*i.e.*, $SpO_2$ as a reflection of the physiological demand), as opposed to altitude elevation (*i.e.*, as an external load index), should perhaps be considered by practitioners (*Millet, Brocherie & Girard, 2016*). In this context, the $SpO_2$ to $FiO_2$ ratio could be used to assess an individual's response to hypoxia, offering a practical method to categorize athletes physiologically and customize $FiO_2$ levels accordingly during altitude training camps (*Soo et al., 2020*).

## Individual variability

Even when athletes underwent identical HiHiLo or LoHi procedures, with the same coach and similar training programs, equivocal results were observed in our study. Some individuals improved their sea level 5-km running time and/or aerobic capacity, while others experienced no change, and some even declined. This considerable within-athlete variability suggests a shift from seeking universal 'best' answers to exploring individual-specific altitude training solutions. Our study highlights that some athletes benefited from relatively moderate hypoxic doses, while others did not, prompting questions about determining the minimum effective dose for each athlete to achieve meaningful gains (*Hauser et al., 2017*). Due to the lack of pre-screening for iron status in athletes, we cannot exclude the possibility that hematological adaptations, and consequently performance benefits, resulting from altitude camps were compromised by insufficient iron availability (*Burtscher et al., 2018*). Another issue is that even individual athletes do not always respond similarly when embarking on the same altitude camp, reinforcing the importance of contextual variables affecting an athlete's response to each altitude training camp. For instance, in the study by *Nummela et al. (2021)*, among the fifteen endurance athletes who participated in altitude training interventions (1,350–2,500 m) at least twice, 27% consistently had positive total hemoglobin mass responses, 13% showed only negative responses, and 60% exhibited both positive and negative responses.

While our study only assessed performance immediately after the camp, it is important to acknowledge the limitation of not conducting a post-intervention assessment in the weeks following altitude training. Postponing the evaluation to several weeks after an altitude camp may reveal performance gains in athletes who initially did not respond or

enhance gains in others (*McLean et al., 2013*). In support, larger 3-km performance enhancements after an 18-d 'live high-train low' altitude camp occurred for athletes training using a natural compared to an artificial hypoxic stimulus 3 weeks after return to sea level, while performance was not elevated above baseline levels immediately after the camp (*Saugy et al., 2014*). In future studies, individualization of the 'altitude dose' would be required to reduce inter-individual variability both during (*i.e.*, training load management) and after (*i.e.*, timing the return to competition) the intervention, which could be aided by EIAH and/or HRV monitoring.

## Exercise-induced arterial hypoxemia

During the incremental test, $SpO_2$ values measured both at a fixed submaximal intensity of 240 W and near exhaustion remained unchanged from pre- to post-intervention in either group. Although the use of pulse oximetry to study EIAH has been questioned (*Durand & Raberin, 2021*), this method has demonstrated high precision, reproducibility, and validity for oxygen saturation above 75% when compared to oxygen saturation measured from arterial blood gases during exercise in normoxia (*Mollard et al., 2010*). In our group of well-trained runners with a $\dot{V}O_{2peak}$ averaging ~65 ml.min$^{-1}$.kg$^{-1}$, the magnitude of EIAH could be classified as '*moderate*' according *Dempsey & Wagner (1999)*, with lowest average $SpO_2$ values ranging 93–88%. Reportedly, two-third of male athletes (*n* = 79) with a $\dot{V}O_{2peak}$ > 68 ml.min$^{-1}$.kg$^{-1}$ displayed EIAH during a progressive-grade treadmill exercise test to exhaustion (*Constantini et al., 2017*). While unchanged EIAH responses occurred when participants were tested near sea level, it cannot be ruled out that improved $SpO_2$ responses would have occurred under hypoxic conditions. In support, *Ventura et al. (2003)* reported a ~3% significant increase in $SpO_2$ after 6 weeks of high-intensity endurance training in hypoxia, but not normoxia, during the post-training incremental test at 3,200 m despite no changes occurring during the same test performed under sea-level conditions.

## Heart rate variability

Although time-domain indices of HRV response to acute hypoxia exposure have been largely studied (*Wille et al., 2012*), little is known about adaptations arising from an altitude camp. In our study, cardiac autonomic activity in response to a 5-km time trial, as inferred from the amount of variability in measurements of the inter-beat interval (LnRMSSD), remained unchanged from pre- to post-tests in both groups (also probably due to our small sample size). This suggests that either HiHiLo or LoHi had no effect on the activity of the parasympathetic system (*Kleiger, Stein & Bigger, 2005*). Because LnRMSSD is not very sensitive to respiratory variations (*Hill et al., 2009*), the fact that unchanged values occurred in the 4-min post-exercise period (with also progressively lower heart rates and longer RR intervals) was not surprising. Perhaps HRV metrics can be used as a diagnostic value for fatigue monitoring during the course of an altitude camp rather than a measure of physiological adaptation from pre-to-post the intervention. In adolescent runners participating in a 2-week altitude training camp, a superior increase in $\dot{V}O_{2peak}$ and subsequent race performance were observed using HRV-guided compared to conventional training methods (*Bahenský & Grosicki, 2021*). Future studies evaluating

training responsiveness should also compare short-duration time-domain HRV indices (*i.e.*, RMSSD) to more time-intensive cardiac autonomic responses to the shifting of body positions captured using HRV frequency-domain parameters.

## Hematological parameters

We found moderate improvement in hematological parameters, with no difference between the two groups. Minimal changes in blood parameters for LoHi are in line with previous studies (*Czuba et al., 2011*) and would likely support non-enhanced erythropoiesis after this form of altitude training. The lack of clear change for all blood parameters in response to HiHiLo was unexpected, as several altitude training studies using a comparable approach (*Czuba et al., 2014*; *Stray-Gundersen, Chapman & Levine, 2001*), but not all (*Chapman, Stray-Gundersen & Levine, 1998*), have reported positive hematological adaptations. As discussed above, the 'altitude dose' (*i.e.*, elevation, duration and daily exposure) may have not been sufficient to invoke robust HIF-1α-related downstream adaptations. Perhaps short daily (<10–12 h), or at the camp level (<200 h), hypoxic exposure time could be compensated by an increased level of hypoxia (*i.e.*, simulated altitude ≥3,500 m (at least during day time), which may maximize haematological responses). Regardless, the absence of meaningful improvements in hematological parameters is not surprising, given that $\dot{V}O_{2peak}$ was not improved with either HiHiLo or LoHi.

## Limitations and additional considerations

Several methodological and logistical limitations must be emphasized. The lack of a significant performance improvement, coupled with unchanged time-domain HRV and EIAH responses in our study, may be partly attributed to the relatively small sample size of sixteen participants. One challenge for research-embedded altitude training camps, as conducted here, is therefore to recruit a sufficiently large number of participants without compromising the validity of findings. While we did not perform an *a priori* sample size calculation, estimating an appropriate sample size for altitude training studies poses challenges as it involves determining the main parameter of interest while considering expected power and effect size. Relying on a single metric may not fully characterize the multi-faceted approach to monitoring athletes in altitude training camp environments (*Saw, Halson & Mujika, 2018*). No control group (*i.e.*, without altitude exposure while sleeping and training) was used, which makes our performance results harder to interpret. Because training load could not be quantified, whether in the lead-in period or during the actual training camp, the effect of the training *per se* could not be evaluated. However, groups were closely matched in terms of performance levels after pre-tests, while all runners also followed the same training program designed by the head coach with the exception of altitude exposure at night. Importantly, nutritional-hydrational status of the athletes during blood tests, well-being during training sessions and sleep quality were not monitored, while total hemoglobin mass could not be measured. Additionally, in this runner cohort, training sessions included a cycling component, and the incremental test was conducted on a cycle ergometer, potentially influencing $\dot{V}O_{2peak}$ determination. These

experimental choices, driven by coach recommendations, aimed to reduce the total amount of running during the altitude training camp (*i.e.*, intensified training period), ensure safety during exhaustive testing, and create a controlled environment for $SpO_2$ measurement. Finally, other forms of 'live high-train low and high' routines could be considered owing to the absence of performance improvement reported here. For instance, utilizing normobaric hypoxic residence for 14 days in combination with repeated maximal-intensity hypoxic exercise conferred a repeated-sprint ability advantage in team sports (*Brocherie et al., 2015*), along with upregulated skeletal muscle molecular adaptations (*Brocherie et al., 2018*), compared to 'live high-train low' or 'live low-train low' groups.

## CONCLUSION

Well-trained runners undertaking base training at moderate simulated altitude for 3 weeks with simulated residence had no physical performance gains compared with no simulated residence, and there were no changes in time-domain HRV and EIAH responses.

### Contribution to the field statement

Athletes are regularly using different altitude training modalities to obtain physiological adaptations and performance benefits. Historically, altitude training emerged in the 1960s and was limited to the 'live high-train high' method for athletes looking for increasing their convective oxygen transport capacity. Other popular altitude training approaches include 'live high-train low' and 'live low-train high', or a combination of both. A promising mixed-method regimen, that may allow for better maintenance of training intensity in reference to 'live high-train low', is known as 'live high-base train high-interval train low'. The aim of this study was to determine the effect of 3-week simulated 'live high-base train high-interval train low' altitude training, compared to 'live low-train high', on changes in physical performance, selected hematological variables as well as heart rate variability and exercise-induced arterial hypoxemia responses immediately after return from the camp. Peak oxygen uptake and running performance along with physiological indices did not differ from pre- to post-tests in either group. In conclusion, well-trained runners undertaking base training at moderate simulated altitude for 3 weeks with or without hypoxic residence had no performance improvement, also with unchanged time-domain heart rate variability and exercise-induced arterial hypoxemia responses.

## ACKNOWLEDGEMENTS

We thank all the participants and coaches who volunteered and contributed to this study. We also thank Zhenyu Liu for his assistant.

### Funding

This research received funding from China Institute of Sport and Health Sciences, Beijing Sport University and Beijing Hundred Thousand Talents Project (2019A15). The funders

had no role in study design, data collection and analysis, decision to publish, or preparation of the manuscript.

### Grant Disclosures
The following grant information was disclosed by the authors:
China Institute of Sport and Health Sciences, Beijing Sport University and Beijing Hundred Thousand Talents Project: 2019A15.

### Competing Interests
The authors declare that they have no competing interests.

### Author Contributions
- Longyan Yi conceived and designed the experiments, analyzed the data, prepared figures and/or tables, authored or reviewed drafts of the article, and approved the final draft.
- Jian Wu performed the experiments, authored or reviewed drafts of the article, and approved the final draft.
- Bing Yan performed the experiments, authored or reviewed drafts of the article, and approved the final draft.
- Yang Wang performed the experiments, authored or reviewed drafts of the article, and approved the final draft.
- Menghui Zou performed the experiments, authored or reviewed drafts of the article, and approved the final draft.
- Yimin Zhang performed the experiments, authored or reviewed drafts of the article, and approved the final draft.
- Feifei Li conceived and designed the experiments, analyzed the data, prepared figures and/or tables, authored or reviewed drafts of the article, and approved the final draft.
- Junqiang Qiu conceived and designed the experiments, analyzed the data, authored or reviewed drafts of the article, and approved the final draft.
- Olivier Girard analyzed the data, prepared figures and/or tables, authored or reviewed drafts of the article, and approved the final draft.

### Human Ethics
The following information was supplied relating to ethical approvals (*i.e.*, approving body and any reference numbers):
The study was approved by Sport Science Experimental Ethics Committee for Human Subjects of Beijing Sport University (Reference No. 2022012H).

### Data Availability
The raw measurements are available in the Supplemental Files.

### Supplemental Information
Supplemental information for this article can be found online at http://dx.doi.org/10.7717/peerj.17166#supplemental-information.

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
