# Peer review of "Effects of three weeks base training at moderate simulated altitude with or without hypoxic residence on exercise capacity and physiological adaptations in well-trained male runners"

_PeerJ, doi:10.7717/peerj.17166_

## Round 0.1 · original submission · Major Revisions

All reviewers found merit in your manuscript but had suggestions for improvement. In addition, the authors should be careful with discussions of "responders/non-responders" unless it can confidently be stated that someone is responding differently. If this cannot be done, please remove that from the manuscript (see: https://physoc.onlinelibrary.wiley.com/doi/full/10.1113/EP085070)

Reviewer 1 ·

Basic reporting

In their manuscript entitled „Effects of three weeks base training at moderate simulated altitude with or without hypoxic residence on exercise capacity and physiological adaptations in well-trained male runners” the authors investigated the effects of HiHiLo and LoHi on VO2max, exercise performance, HRV during exercise and the recovery phase and hematological parameters (tHb mass was not determined, which was considered a limitation by the authors themselves). No effects of the two training regimes on those parameters were reported.

Experimental design

Methods:
Participants:
Did the authors perform a sample size calculation? If so, on what data was it based?

Study design:
Line 156-159: I would suggest creating a table of baseline characteristics for the two groups that includes, for example, HRmax, VO2max, Wattmax, best 5000m time, training experience, etc., in addition to age, body mass, and height.

Line 160: The authors could specify here that the tests were performed on two days “immediately” after the training program.

Altitude exposure:
Line 169 and 173: Why did the authors choose to stay at an altitude of 2800 and for (only) 10 hours per day? The recommendations seem to be slightly different (Girard et al. https://doi.org/10.1123/ijspp.2022-0501).

170-171: How was the CO2 level controlled?
Line 177: Exercise intensity was set at ~170 bpm. I wonder why the authors did not select a relative intensity (i.e., %HRmax) for training intensity prescription? An absolute HR of 170 bpm could have resulted in very different training loads. What was HRmax (SD) during testing?
Line 182: What other training was done besides altitude training? This was not described in the manuscript.

Incremental test:
Line 184-194: Why did the authors decide to perform a cycling test protocol for runners? How might this have affected the VO2max determination? The authors set criteria for reaching exhaustion, but I could not find in the manuscript if the criteria were actually met. Please also consider VO2max vs. VO2peak.

Athletic performance:
Line 196-198: the authors could include here that testing was performed before and after the intervention. Did the authors record heart rate or RPE during the runs? In my opinion this could be valuable information.

Heart rate variability:
Line 207-213: The authors could include a ref supporting their approach.

Validity of the findings

no comment

Additional comments

Key point 3, line 58-72: Key point 3 seems to me to be a summary rather than a key point. I would suggest revising this point.

Abstract:
Line 78: The authors could add “young” to the description of the participants.
Line 87-92: The authors could report some values together with the p-value so that the effects are already visible in the abstract.

Introduction:
Line 124-126: If I am not mistaken, the cited reference (Chapman et al. 2011) only addresses EIAH and not HRV. While the hypothesis to investigate the effects of EIAH seems sound the authors should address the rational for including HRV measurements in more detail.

Results:
Have there been changes in body weight? Could this have influenced results?


Discussion:
Individual variability:
In addition to the responders and non-responders issue, it might be interesting to include information on the repeatability of altitude training (doi: 10.1111/sms.13804).

Exercised-induced arterial hypoxemia:
Why did the authors expect exercised-induced arterial hypoxemia to occur already at 240 W? In this regard I wonder how the authors explain the outliers in Figure 1 (VO2 and SpO2at 240W). VO2 at 240 W of <40 ml/min/kg or SpO2 of < 85% in normoxia seems excessively low. In addition, the variability of 5-km time seems to be higher in the LoHI group than in the HiHiLo group.

Blood parameters:
Line 330-341: The limited significance of the measured hematological markers could be addressed here too.

·

Basic reporting

No comment

Experimental design

Most relevant limitations have been mentioned, especially the lack of a control group without any hypoxia exposure.
In addition, the sample size is relatively small. Did the authors perform any sample size and power calculations (at least a-posteriori)?
Moreover, the authors state that “participants were near sea-level residents and had not travelled to altitude (>1500 m) in the six months prior to the study”; did participants use simulated altitude training prior to the study?
Did the authors check individual iron status? Lack of iron may have prevented from sufficient erythropoiesis and increase in hemoglobin mass (e.g., see PMID: 26183475, 30425646).

Validity of the findings

The paper would benefit from a more detailed discussion of individual factors potentially contributing to different responses to altitude training.
It is always a bit surprising when athletes do not show any performance improvements after a 3-wk training camp. The authors may elaborate a bit on that.
Conclusions: you may delete "extra" in the first sentence as there were no performance gains within both groups.

Additional comments

The presented outcomes are not really surprising but highly relevant (practically and scientifically) as individual responses to altitude/hypoxia training are still given too less attention!

Nice study!

Reviewer 3 ·

Basic reporting

- Line 68-69: Consider changing “before to after” description to “pre- to post”, as in accordance to the rest of the paper.
- Line 105: To better explain the foundations of altitude training and the physiological mechanisms underlying the purpose of pursuing such a training paradigm, I recommend further detail in explaining “fall in partial pressure due to hypoxia”. Is this in regard to the atmospheric pressure or the partial pressure of O2 or CO2 in the human body?
- Line 122: Consider changing “controversy” to explaining more along the lines that the evidence is conflicting or unclear.
- Line 270: Consider further explaining the “attractive” metric of using SpO2 as the metric for physiological demand rather than external load of dose of hypoxia from Millet (2016).
- Line 273: “programmes” may be changed to “programs”.
- Lines 272-278: Sentence structure and grammar could be improved to better explain the reported results.
- Line 296: Consider adding an article, “the” when introducing the observation regarding “the incremental test”.
- Line 343: Consider adding a section that explicitly lists and defines the limitations of the paper, including intermittently introduced limitations from the previous paragraphs of the discussion section. This may allow a greater resolution to the discussion of the paper before the conclusion.
- Raw data shared should be commended. However, consider adding/ explaining the heading of variable names so that it is clear what the data is referring to.
- Overall, the language is very professional throughout the manuscript.

Experimental design

- Line 255: Robertson et al. (2010) demonstrated contrasting results to the current paper; they utilized a similar 3-week timetable, but the training modality during the intervention was exclusively running – the more natural training stimulus for the participants compared to the 30 min cycle/30 min running split in the current paper. Please explain the justification for this intervention in the current paper.
- The use of the cycle ergometer test is clearly explained from past literature. However, why could a max test on the treadmill not be used for these runners? Providing justification for this approach would be valuable.
- It was not indicated as to why blood lactate samples were not collected. Is there a justification for this decision? Previous work by Bahensky et al. (2020) demonstrated significant changes regarding blood lactate levels and running velocity from altitude training, which may be worth considering.

Validity of the findings

- Lines 311-321: A limitation was mentioned evaluating HRV post 5km trial (which demonstrated no significant changes between groups) compared to a HRV metric as a diagnostic value assessing recovery/fatigue from training stimuli during the intervention. The findings are interpreted as “not surprising”. The supported literature that demonstrated contrasting results (Hamlin, 2011) investigated a “live-moderate, train-low” sample, which may not be a suitable observation to replicate in this sample. Furthermore, the reference is not in the references section – if these findings related to this HRV metric at a higher altitude are not surprising, would it be valuable to readjust the approach for future work regarding HRV to determine if the two different groups have different fatigue/recovery responses? Could there be a stronger justification for the approaches used in this paper?
- The general approach described in the study design not fully make it clear so to why certain approaches were taken; i.e., there is not adequate links from the literature introduced to explain the methods and reported findings regarding HRV. Consider introducing the literature that was placed in the results section (lines 312-319) in the introduction section to better lay out the purpose of this approach.
- Another main limitation was the exclusion of a control group compared to the two intervention groups. Although the purpose of the paper was to assess the two different intervention groups, a future project building on these methods should include a control group to better evaluate the results as it relates to the effectiveness of altitude training among elite distance runners.

Additional comments

The authors introduced their research question and hypotheses assessing the two different interventions of altitude training, HiHiLo and LoHi. A strength of the study was employing these 2 distinct intervention groups while evaluating work capacity, running performance, and blood parameters. Some of the main areas of improvement include a more thorough review of sentence structure and grammar throughout the paper, a more thorough description of the study design that includes a more direct relation of previous literature as it relates to the current study design, and the inclusion of a control group. Furthermore, the methodology of HRV should be reevaluated. The null findings of the paper further demonstrate the need to evaluate altitude training in the distance running community. Future work including a control group would further emphasize the effectiveness of the different modalities of altitude training. Overall, I feel that with these considerations and revisions in mind, this paper could make a solid contribution to the literature. The authors should be commended for all their hard work that was put into this project.

---

## Round 0.2 · accepted · Accept

Reviewers agree that this manuscript is ready for publication.

Reviewer 1 ·

Basic reporting

Thank you for responding to all my comments. I have no further questions.

Experimental design

I have no further questions.

Validity of the findings

I have no further questions.

Additional comments

I have no further questions.

·

Basic reporting

No comment

Experimental design

No comment

Validity of the findings

No comment

Additional comments

I appreciate the efforts made by the authors in revising their manuscript.
Well done! No further comments.

Reviewer 3 ·

Basic reporting

no comment

Experimental design

no comment

Validity of the findings

no comment

Additional comments

After a thorough review and addressing of comments from all reviewers, the authors should be commended for their hard work for developing a suitable manuscript.